# Influence of the Catecholamine Syringe Changeover Method on Patients’ Blood Pressure Variability: A Single-Center Retrospective Study

**DOI:** 10.3390/nursrep15100345

**Published:** 2025-09-23

**Authors:** Yuta Niitsu, Takumi Tsuchida, Ryuta Sato, Juna Shintaku, Koichi Iwasa

**Affiliations:** 1Critical Care Nurse Center, Hokkaido University Hospital, Sapporo 060-8648, Japan; 2Department of Emergency Medicine, Hokkaido University Hospital, Sapporo 060-8648, Japan

**Keywords:** critical care, catecholamine syringe exchange, intensive care unit, syringe pump, shock, vasopressor

## Abstract

**Background/Objectives**: In Japan, evidence on catecholamine syringe exchange methods is limited, with practices varying across facilities and individuals. In this study, we aimed to determine the effect of the catecholamine syringe exchange method on blood pressure variability in intensive care unit patients. **Methods**: We retrospectively analyzed 119 patients (308 syringe exchanges) who underwent catecholamine syringe exchange between 1 April 2020 and 31 March 2022. Patient characteristics for the double-pumping changeover (DPC) and quick syringe changeover (QC) groups were matched and compared using propensity scores. A sub-analysis focused on patients with severe shock with systolic blood pressures ≤ 90 mmHg. Logistic regression analysis was used to examine factors influencing blood pressure variability during the catecholamine syringe changeover. **Results**: Neither propensity score matching nor the sub-analysis for patients with shock revealed significant differences in the coefficient of variation or absolute systolic/diastolic/mean blood pressure within 15 min of syringe exchange in the two groups. Logistic regression revealed that age was the sole risk factor affecting blood pressure variability during syringe changeover (odds ratio: 1.018, 95% confidence interval: 1.001–1.036), while syringe changeover methods did not contribute to circulating variability (odds ratio: 1.186, 95% confidence interval: 0.672–2.092). **Conclusions**: Differences between the DPC and QC methods did not significantly affect blood pressure variability during catecholamine syringe changeovers. However, in older adult patients, catecholamine syringe changeover may be more likely to cause blood pressure variability.

## 1. Introduction

Catecholamine preparations are commonly administered using syringe infusion pumps; however, interruptions during syringe changeover can lead to blood pressure fluctuations. Moreover, several incidents related to syringe infusion pump use have been attributed to drug changeovers [1]. Because the syringes used in these pumps have a relatively small capacity, they require frequent replacement, which may also cause instability in circulatory dynamics [2]. In the literature, syringe changeover is described as a particularly dangerous procedure during the administration of vasoactive agents, as various technical and physiological factors may contribute to backflow or unintended bolus formation [3]. Representative methods for syringe exchange include the single-syringe, double (parallel), and quick exchange methods. In the single-syringe method, infusion is temporarily interrupted for syringe replacement and motor gear alignment, making this approach more susceptible to blood pressure fluctuations [4]. To minimize hemodynamic instability during syringe exchange, the parallel or quick exchange method is generally preferred [5].

The parallel exchange method uses two syringe infusion pumps to overlap the infusion during the exchange; however, if the operator is inexperienced, the pump may stop for approximately 15–30 s while the flow rate is manually adjusted [6]. Consequently, the parallel exchange method poses a circulatory fluctuation risk owing to differences in the nursing staff’s experience and skills. It has been reported that systolic blood pressure can change by up to 50 mmHg with this method [7].

The quick exchange method also uses two pumps, requiring switching between them via a three-way stopcock, resulting in the completion of the process in approximately 5 min. Reducing the operation time is believed to be the most effective way to maintain blood pressure during the syringe exchange of catecholamine preparations [8]. As the quick exchange method can be completed in approximately 5 min, it is expected to result in fewer blood pressure fluctuations. Experimental studies using flow measurement devices have shown that the quick-exchange method has less flow rate error than the parallel exchange method [9].

Previous randomized controlled trials conducted in Europe on human participants found that compared with the parallel exchange method, the quick exchange method is less likely to cause a decrease in mean blood pressure exceeding 15 mmHg [10]. However, it has also been reported that 12.4% of patients experience a decrease of >20% in mean blood pressure with the quick exchange method [2], suggesting that the quick exchange method’s impact on blood pressure remains inconclusive. No large-scale studies on syringe exchange methods have been conducted in Japan, and no method has been identified or recommended as optimal for Japanese patients and environments, resulting in varying practices across facilities. Furthermore, unlike in Europe, automated changeover used in previous studies has not become prevalent in Japan [10], and due to racial differences, evidence specific to Japan does not currently exist.

Given this background, we conducted a retrospective study to clarify the impact of syringe exchange methods on blood pressure variability by comparing the parallel and quick exchange methods and identifying risk factors contributing to fluctuations.

## 2. Materials and Methods

We conducted a retrospective study on patients admitted to an Emergency and Critical Care Center (between 1 April 2020, and 31 March 2022) who underwent syringe exchange for catecholamine preparations under invasive arterial pressure measurement. The target drugs included catecholamine preparations (adrenaline, noradrenaline, dobutamine, and dopamine), vasopressin, milrinone, and normal saline solution as a diluent for catecholamine preparations. Multiple data points were collected from each patient.

The exclusion criteria were based on factors potentially affecting hemodynamics during the measurement period. The list of exclusions are provided in Appendix A. The specific criteria are as follows: (a) changes in catecholamine preparation infusion rate; (b) initiation or changes in sedative drug infusion; (c) administration of antipyretics, diuretics, steroids, or electrolytes; (d) bolus fluid administration or blood transfusion; (e) initiation of continuous dialysis, circuit exchange, or changes in fluid removal; (f) initiation or changes in mechanical circulatory support; (g) changes in ventilator settings; (h) body positioning or hygiene care; (i) airway secretion suction; (j) therapeutic interventions; (k) rehabilitation; (l) movement or transfer within the bed; and (m) delirium symptoms. We also excluded cases where invasive arterial pressure measurement was not possible due to (n) leaving the room for tests, (o) arterial line issues, (p) missing measurement data, (q) blood gas analysis, or (r) non-insertion of the arterial pressure line.

The target patients were classified into the parallel and quick exchange groups, and the evaluation items were compared between both groups. Additionally, propensity score matching was conducted by narrowing down the target drugs to noradrenaline only and adjusting for patient backgrounds between both groups. The covariates included age, sex, primary disease, medical history, number of drugs through the exchange route, exchange drug infusion rate (γ), and coefficient of variation in blood pressure 30 min before syringe exchange. The caliper was set at 0.1. A subgroup analysis focused on patients with severe shock who were unable to maintain systolic blood pressure above 90 mmHg under catecholamine administration.

Additionally, to identify risk factors for blood pressure fluctuations, logistic regression was performed on the entire cohort. Cases in which the coefficient of variation in mean blood pressure 15 min after syringe exchange exceeded the median were defined as “blood pressure fluctuation present,” while those with values below the median were defined as “blood pressure fluctuation absent.” Independent variables included age, sex, primary disease (infection, post-cardiac arrest, burns, or hypothermia), medical history (hypertension, diabetes, or heart disease), number of catecholamines used (single or two or more agents), infusion rate of the exchanged drug, and syringe exchange method.

Study outcomes were evaluated using the coefficient of variation in blood pressure (standard deviation/mean value) and the absolute value of blood pressure changes from immediately after syringe exchange to 15 min later, a period during which flow rate changes in the parallel exchange are large and less influenced by factors other than syringe exchange.

Collected data included patient age, sex, primary disease, medical history, type of exchanged drug, number of drugs in the exchange route, total flow rate of the exchange route at the time of exchange, infusion rate of the exchanged drug, dosage of the exchanged drug (µg/kg/min = γ), duration of parallel exchange, and time of exchange. Blood pressure data consisted of systolic and diastolic blood pressure values recorded at 1 min intervals, averaged every 5 min. Blood pressure was measured 30 min before and 25 min after syringe exchange.

The actual procedure for the exchange method was as follows: Two syringe infusion pumps were used in the parallel exchange method, with the infusion rates of the old and new syringes adjusted by increasing or decreasing them to ensure that the total flow rate remained constant at all times. The operator monitored the invasive arterial pressure values and performed the procedure in such a way as to minimize blood pressure fluctuations; therefore, the timing of the rate adjustments was not fixed. The duration from start to finish also varied, with an average of approximately 25 min.

In the quick exchange method, the new syringe was preset to 5 mL/h and allowed to flow freely for 5 min. Subsequently, the infusion rate of the new syringe was adjusted to match that of the old syringe, and the new syringe was connected to the line of the old syringe. The lines of the old and new syringes were switched using a three-way stopcock. Therefore, the syringe stoppage time in the quick exchange method was only a few seconds at most.

At the study facility, the infusion flow rate from the same route was set to ≥2 mL/h to prevent dosage fluctuations at low infusion rates. The syringe exchange method was left to the judgment of individual nurses, as no specific protocol was in place.

Statistical tests between the two groups were performed using Student’s t-test for normally distributed data and the Mann–Whitney U test for non-normally distributed data. The results are presented as the median (interquartile range). Pearson’s chi-squared test or Fisher’s exact test was used to test the ratios. All analyses were conducted using IBM SPSS (version 25, IBM Japan, Tokyo, Japan), with a significance level set at *p* < 0.05.

This study was approved by the Hokkaido University Hospital Ethics Committee (approval number: 22-1). In accordance with the Personal Information Protection Law, the information related to the study was managed by replacing it with different numbers (research IDs) so that it could not be immediately linked to specific individuals. Since this study did not involve the use of samples obtained from human participants, obtaining informed consent was not required under the “Ethical Guidelines for Medical and Health Research Involving Human Subjects” (Ministry of Education, Culture, Sports, Science, and Technology and Ministry of Health, Labour and Welfare Notification No. 3 of 2014), and this process was omitted. The study details and purpose were posted on the University Hospital website, allowing participants to opt out.

## 3. Results

From an initial 674 exchanges, we excluded 291 cases involving circulatory-affecting procedures and 75 cases where obtaining invasive arterial pressure measurement was not possible. Of the remaining 308 cases, 120 were included in the parallel exchange group and 188 in the quick exchange group, involving 119 total patients (Figure 1).

### 3.1. Unadjusted Comparison Between the Parallel and Quick Exchange Groups

The patient backgrounds for both groups are shown in Table 1. The parallel exchange group included 120 cases, with 79 male patients (65.8%) and a median age of 70.0 years. The quick exchange group included 188 cases, with 106 male patients (56.4%) and a median age of 68.5 years. No significant differences were observed between the groups in terms of the background factors such as sex, primary disease, diabetes history, number of drugs in the exchange route, total flow rate, infusion rate, or drug dosage. However, significant differences were observed in age (*p* = 0.005), hypertension (*p* = 0.009), heart disease (*p* = 0.001), and the number of catecholamines used (*p* = 0.024) (Table 1). No significant differences were found between the groups in the coefficient of variation or absolute values of systolic, diastolic, or mean blood pressure within 15 min after syringe exchange (Table 2).

### 3.2. Comparison Between the Parallel Exchange Group and the Quick Exchange Group After Propensity Score Matching

In the overall comparison, there were differences in patient backgrounds; therefore, a comparison was made using propensity score matching, limiting the analysis to the frequently used drug noradrenaline. Patient backgrounds after matching are shown in Appendix A. The parallel exchange group included 49 men (64.4%) with a median age of 67.5 years, whereas the quick exchange group included 49 men (64.4%) with a median age of 68.0 years. No significant differences were observed between the two groups in any variables, including age, sex, or medical history. The comparison results after propensity score matching are presented in Table 3. There were no significant differences in the coefficients of variation or absolute values of systolic/diastolic/mean blood pressure between the two groups up to 15 min after syringe exchange.

### 3.3. Syringe Exchange in Patients with Severe Shock (Sub-Analysis)

Syringe exchanges in patients with a systolic blood pressure ≤ 90 mmHg in the parallel and quick exchange groups included 19 and 26 cases, respectively. Significant differences were found in primary disease (infection, *p* = 0.037; burns, *p* = 0.014) and medical history (cardiovascular disease, *p* = 0.044) (Appendix A). Although patient backgrounds were not adjusted, no significant differences were found in the coefficients of variation or absolute values of systolic/diastolic/mean blood pressure between the two groups up to 15 min after syringe exchange in the subanalysis (Appendix A).

### 3.4. Risk Factors for Syringe Exchange of Catecholamine Drugs

The primary outcome of this analysis was the presence of blood pressure fluctuation, defined as a coefficient of variation exceeding the median within 15 min of syringe exchange. A total of 154 patients (half of the study population) were classified as “presence of blood pressure fluctuation.” Logistic regression was performed using five explanatory variables: age, sex, medical history, number of drugs in the exchange route, dose of the exchange drug, and exchange method (parallel or quick exchange). The only factor influencing blood pressure fluctuations within 15 min after syringe exchange was age (odds ratio: 1.018, 95% confidence interval: 1.001–1.036, Figure 2). The syringe exchange method did not contribute to blood pressure fluctuations (odds ratio: 1.186, 95% confidence interval: 0.672–2.092).

## 4. Discussion

In this study, no significant differences were observed between the parallel and quick exchange groups regarding the effects of catecholamine drugs on blood pressure fluctuations during syringe exchange. Logistic regression analysis showed that age was the only factor influencing blood pressure fluctuations during the syringe exchange of catecholamine drugs.

It is well recognized that syringe exchange is a challenging procedure, particularly during vasoactive drug infusion. Elli et al. emphasized that factors such as syringe pump position relative to the patient and the use of needle-free connectors influence the risk of backflow or inadvertent bolus [3]. Consistent with this, a recent bench-top study simulating cardiac output demonstrated that central venous pressure and vertical pump position significantly affected bolus or backflow during syringe pump changeover [11]. These findings reinforce the notion that both procedural and technical factors must be carefully controlled to minimize hemodynamic variability.

A Japanese observational study reported that patients with blood pressure > 20 mmHg above target showed syringe exchange usage rates of 33.3% for the double method and 20.1% for the quick method. Conversely, for patients with blood pressure > 20 mmHg below target, the rates were 83.3% for the double method and 11.8% for the quick method, suggesting lower use of the quick method in such cases [5]. However, our study showed that the quick exchange method was chosen more often for patients with cardiovascular diseases, and more concomitant drugs were used (Table 1). Additionally, a sub-analysis was conducted to clarify the impact of syringe exchange on the hemodynamics of severely ill patients, focusing on syringe exchanges performed in patients with severe shock; however, no influence of the syringe exchange method on blood pressure fluctuations was found.

Previous experimental studies using flow meters to compare the parallel and quick exchange methods found no differences in flow rate errors between the groups [8]. Similarly, this clinical study targeting patients requiring catecholamines found no differences between the groups. Moreover, the quick exchange method is less likely to cause a mean blood pressure decrease of >15 mmHg than the parallel exchange method [9]. In this study, no significant differences were observed in the absolute values between the two groups. While syringe infusion pumps are known to take some time to stabilize at the set flow rate after operation [4], the quick exchange method involves pre-driving a new syringe, possibly reducing drug delivery delays, which may have resulted in similar blood pressure fluctuations as the parallel exchange method.

Regardless of the syringe exchange method, it has been reported that the frequency of mean blood pressure decreases is lower with high-flow noradrenaline syringe exchanges than with low-flow exchanges [2]. However, our study found no significant association between flow rate and blood pressure fluctuations within 15 min of exchange. Blood pressure fluctuations may occur when the infusion rate of a syringe infusion pump is <1 mL/h [12]. In this study, the infusion rate from the same delivery route was maintained at ≥2 mL/h, which may have influenced the results.

Logistic regression revealed that in older adult patients, catecholamine exchanges were associated with greater circulatory fluctuations. Physiological changes in aging, including enhanced pressor reactivity to noradrenaline via sympathetic alpha-receptor stimulation [13], may explain why even slight flow changes resulted in significant blood pressure fluctuations during syringe exchanges.

In this study, no differences were observed between the effects of parallel and quick exchange methods on blood pressure fluctuations. As previously discussed, the parallel method is more complex, with increased risk of circulatory fluctuations depending on the nurse’s skill and experience. However, the quick exchange method is relatively simple, and the time to complete the exchange is shorter. Therefore, based on these results, the quick exchange method may be preferable for catecholamine syringe exchange. However, this study evaluated blood pressure fluctuations when syringe exchanges were conducted ideally and without complications, and it did not assess the success rate of the exchange procedures. Therefore, further evidence regarding the exchange methods is necessary.

### 4.1. Limitations

This study had some limitations. First, this was a single-center retrospective study, and there may be unmeasured confounding factors as well as the possibility of patient selection bias. Second, the sample size was relatively small, particularly in cases of severe shock, where the number of exchanges was limited, potentially making the statistical results unstable. Lastly, information regarding the nurses performing the syringe exchanges was unavailable; therefore, factors such as years of experience, exchange techniques, and preferences for certain methods may have influenced the results. The nurse selected the syringe exchange method; therefore, potential selection bias may be present.

### 4.2. Implications for Emergency Nursing

This study demonstrates that neither the double-pumping changeover nor the quick syringe changeover significantly affects blood pressure variability, emphasizing that age, rather than syringe exchange method, is a primary factor influencing variability. Emergency nursing practices should prioritize careful monitoring of older patients during catecholamine syringe exchanges, as age-related factors may increase susceptibility to blood pressure variability.

## 5. Conclusions

No difference was observed in the impact on blood pressure fluctuations between the parallel exchange method and quick exchange method for syringe exchanges of catecholamine drugs. Blood pressure fluctuations are more likely to occur during catecholamine syringe exchange in older adult patients. Future multicenter and prospective studies are warranted to validate these findings.

## Figures and Tables

**Figure 1 nursrep-15-00345-f001:**
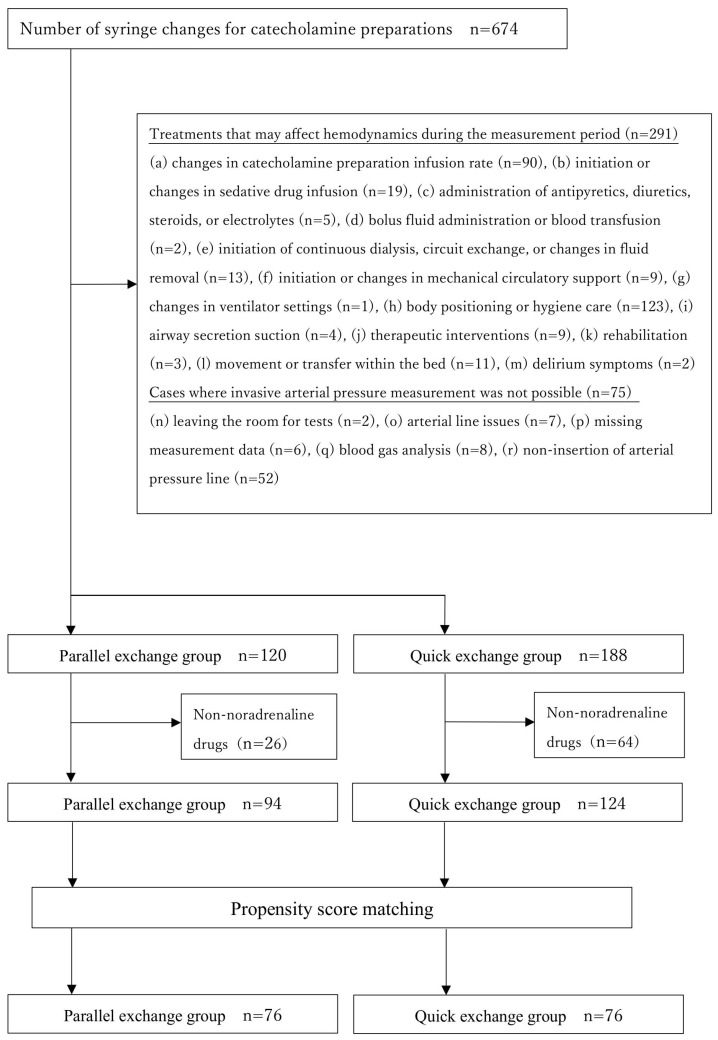
Flowchart of the included syringe exchange case.

**Figure 2 nursrep-15-00345-f002:**
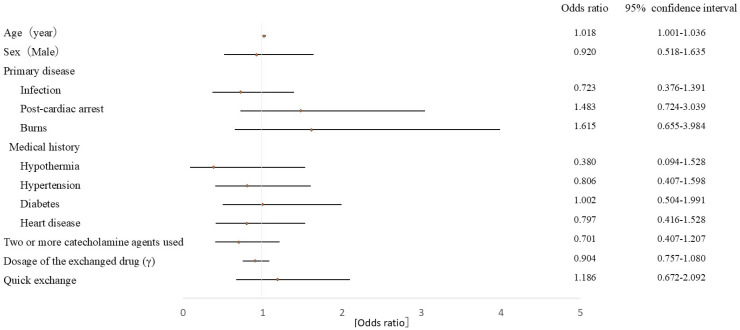
Risk Factors for Syringe Exchange of Catecholamine Preparations (Logistic Regression Analysis Results).

**Table 1 nursrep-15-00345-t001:** Patient Background.

	Parallel Exchange (*n* = 120)	Quick Exchange (*n* = 188)	*p* Value
Age	70.0 (64.8–83.0)	68.5 (50.8–80.3)	0.005
Sex (male, %)	79 (65.8)	106 (56.4)	0.099
Primary disease			
Infection (n, %)	56 (46.7)	70 (37.2)	0.101
Post-cardiac arrest (n, %)	35 (29.2)	59 (31.4)	0.680
Burns (n, %)	23 (19.2)	24 (12.8)	0.128
Hypothermia (n, %)	6 (5.0)	10 (5.3)	0.902
Medical history			
Hypertension (n, %)	44 (36.7)	45 (23.9)	0.009
Diabetes (n, %)	23 (19.2)	54 (28.7)	0.083
Heart disease (n, %)	21 (17.5)	68 (36.2)	0.001
Type of exchanged drug			0.071
Noradrenaline (n, %)	94 (78.3)	124 (66.0)	
Dobutamine (n, %)	11 (9.2)	37 (19.7)	
Vasopressin (n, %)	9 (7.5)	20 (10.6)	
Adrenaline (n, %)	4 (3.3)	2 (1.1)	
Dopamine (n, %)	0 (0.0)	2 (1.1)	
Milrinone (n, %)	0 (0.0)	1 (0.5)	
Normal saline solution to boost catecholamine (n, %)	2 (1.7)	2 (1.1)	
Number of catecholamines used (single, %)	63 (52.5)	57 (30.3)	0.024
Total flow rate of the exchange route at the time of exchange (mL/h)	5.6 (3.6–9.1)	6.4 (4.1–8.25)	0.249
Infusion rate of the exchanged drug (mL/h)	4.15 (2.1–6.1)	4.1 (2.5–5.6)	0.825
Dosage of the exchanged drug (γ)	0.2 (0.1–0.3)	0.2 (0.1–0.5)	0.064
The coefficient of variation in mean blood pressure during the 30 min before syringe exchange	0.029 (0.019–0.050)	0.030 (0.017–0.049)	0.517
The time of exchange (Night, %)	68 (56.7)	96 (51.1)	0.337

This table is shown with n (%) or median (25–75th percentile).

**Table 2 nursrep-15-00345-t002:** Comparison of the Parallel Exchange Group and the Quick Exchange Group.

		Parallel Exchange (*n* = 120)	Quick Exchange (*n* = 188)	*p* Value
Coefficient of variation				
After 5 min	SBP	0.026 (0.013–0.050)	0.024 (0.012–0.046)	0.542
	DBP	0.021 (0.009–0.042)	0.022 (0.009–0.041)	0.901
	MBP	0.024 (0.011–0.049)	0.023 (0.010–0.043)	0.473
After 10 min	SBP	0.037 (0.021–0.067)	0.037 (0.020–0.061)	0.674
	DBP	0.031 (0.017–0.055)	0.034 (0.017–0.053)	0.880
	MBP	0.035 (0.019–0.059)	0.034 (0.018–0.059)	0.589
After 15 min	SBP	0.040 (0.026–0.071)	0.043 (0.024–0.071)	0.661
	DBP	0.038 (0.024–0.057)	0.036 (0.020–0.064)	0.631
	MBP	0.040 (0.024–0.064)	0.039 (0.022–0.067)	0.472
Absolute value of the change				
After 5 min	SBP	−1.0 (−6.0–6.0)	−3.0 (−8.0–2.0)	0.050
	DBP	−1.0 (−3.0–2.0)	−1.0 (−3.0–2.0)	0.978
	MBP	−0.7 (−4.3–3.3)	−1.3 (−4.4–1.3)	0.209
After 10 min	SBP	0.0 (−7.0–5.0)	−2.5 (−7.0–4.0)	0.303
	DBP	0.0 (−3.0–2.0)	0.0 (−3.0–3.0)	0.761
	MBP	0.0 (−4.8–3.0)	−0.7 (−4.4–2.7)	0.683
After 15 min	SBP	−2.0 (−8.0–3.0)	−2.0 (−8.0–4.3)	0.680
	DBP	−1.0 (−3.3–1.0)	0.0 (−3.0–3.0)	0.194
	MBP	−1.3 (−5.0–2.3)	−1.2 (−4.7–3.0)	0.538

This table is shown with median (25–75th percentile). SBP: systolic blood pressure, DBP: diastolic blood pressure, MBP: mean blood pressure.

**Table 3 nursrep-15-00345-t003:** Comparison After Propensity Score Matching.

		Parallel Exchange (*n* = 76)	Quick Exchange (*n* = 76)	*p* Value
Coefficient of variation				
After 5 min	SBP	0.027 (0.013–0.049)	0.035 (0.017–0.055)	0.268
	DBP	0.018 (0.009–0.042)	0.027 (0.010–0.046)	0.152
	MBP	0.024 (0.011–0.044)	0.032 (0.011–0.046)	0.360
After 10 min	SBP	0.041 (0.022–0.064)	0.048 (0.024–0.073)	0.354
	DBP	0.033 (0.015–0.050)	0.037 (0.021–0.054)	0.398
	MBP	0.036 (0.019–0.057)	0.040 (0.020–0.064)	0.454
After 15 min	SBP	0.042 (0.026–0.073)	0.054 (0.028–0.079)	0.560
	DBP	0.040 (0.024–0.056)	0.041 (0.026–0.063)	0.799
	MBP	0.041 (0.024–0.065)	0.046 (0.026–0.073)	0.737
Absolute value of the change				
After 5 min	SBP	−1.0 (−8.0–5.3)	−3.0 (−10.0–4.3)	0.398
	DBP	−1.0 (−3.3–1.3)	0.0 (−3.0–2.3)	0.459
	MBP	−1.0 (−5.0–2.7)	−1.2 (−5.8–2.2)	0.892
After 10 min	SBP	−1.0 (−9.0–4.0)	−1.5 (−8.3–4.0)	0.867
	DBP	0.0 (−3.0–1.0)	0.0 (−3.0–3.0)	0.226
	MBP	0.0 (−5.0–2.0)	−0.3 (−4.8–3.8)	0.538
After 15 min	SBP	−2.0 (−6.0–4.0)	−2.5 (−8.0–5.0)	0.839
	DBP	−1.0 (−4.0–1.3)	0.0 (−3.0–3.0)	0.411
	MBP	−1.3 (−4.4–2.6)	−1.2 (−4.8–3.4)	0.818

This table is shown with median (25–75th percentile). SBP: systolic blood pressure, DBP: diastolic blood pressure, MBP: mean blood pressure.

## Data Availability

The data presented in this study are available on request from the corresponding author (the data are not publicly available due to privacy restrictions).

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
