# Peer review of "Influence of the Catecholamine Syringe Changeover Method on Patients’ Blood Pressure Variability: A Single-Center Retrospective Study"

_nursrep, 2025, doi:10.3390/nursrep15100345_

Round 1
Reviewer 1 Report
Comments and Suggestions for Authors
You can see my comment below.
- Abstract, Introduction, and methods were well written
- In Table 1, the p value was for Type of exchange drug?
- Discussion section is important for result interpretation. Although some results were insignificant, they may be important to increase alertness of the users of syringe pumps, comparing the results of the similar studies.
- The Discussion can further elaborate the factors that help improve the current practice.
- You also need to give suggestions based on your results to facilitate evidence-based practice.
Author Response
Comments 1: In Table 1, the p value was for Type of exchange drug?
Response 1: Thank you for your question. The p-values listed in Table 1 under “Type of exchanged drug” indicate whether there was a difference in the type of drug exchanged. In this study, both Parallel exchange and Quick exchange involved a high proportion of norepinephrine exchange, and no significant difference was observed.
Comments 2: Discussion section is important for result interpretation. Although some results were insignificant, they may be important to increase alertness of the users of syringe pumps, comparing the results of the similar studies. The Discussion can further elaborate the factors that help improve the current practice.
Response 2: We sincerely thank the reviewer for this thoughtful comment and fully agree that elaborating on factors to improve current practice would strengthen the Discussion. However, as there are very few existing studies directly addressing this topic, it was challenging to expand the discussion further while maintaining accuracy and avoiding speculation. We have therefore retained a concise description and emphasized this limitation in the Discussion. We hope this approach will be acceptable.
Comments 3: You also need to give suggestions based on your results to facilitate evidence-based practice.
Response 3: We sincerely thank the reviewer for this valuable comment emphasizing the importance of providing suggestions to facilitate evidence-based practice. We fully agree with this point. To address this, we have already included a section entitled “Implications for Emergency Nursing” in the Discussion, where we summarize the practical applications of our findings. We hope this adequately addresses the reviewer’s concern.
Reviewer 2 Report
Comments and Suggestions for Authors
please see the attachment.

< !--a=1-->< !--a=1-->< !--a=1-->
Author Response
Comments 1: The singular “patient’s” may suggest that only one patient was studied; the plural possessive (“patients’”) or a descriptive form would be more appropriate.
The phrase “catecholamine syringe changeover method” is correct but could read more smoothly with slight restructuring.
Response 1: Thank you for your excellent feedback and kind suggestion regarding the title.
As you pointed out, I have changed the title to “Influence of the catecholamine syringe changeover method on patients’ blood pressure variability: a single-center retrospective study”.
Abstract
Comments 2: The methods and results are clearly stated, but the abstract should explicitly mention that this is a retrospective study to aid interpretation.
Response 2: Thank you for your feedback. We have added a note to the abstract to clarify that this is a retrospective study (line 13).
Introduction
Comments 3: The explanation of the changeover methods (lines 35–47) contains some repetition and could be streamlined for clarity.
Response 3: Thank you for your feedback. We have revised the opening of the introduction to avoid repetition (lines 34 to 44).
Comments 4: The discussion of limited evidence from Japan (lines 62–70) would benefit from a brief note on how the local context may differ from European practice (e.g.,
protocols, staffing ratios).
Response 4: Thank you for your feedback. Taking your opinion into consideration, I have added the phrase “Furthermore, unlike in Europe, automated changeover used in previous studies has not become prevalent in Japan, and due to racial differences, evidence specific to Japan does not currently exist.”” to the latter part of the introduction (lines 67–69).
Materials and Methods
Comments 5: The list of exclusions (lines 82–93) is comprehensive but could be presented in a table or structured list for easier reading.
Response 5:Thank you for pointing out, we have created the exclusion table as Table S1.
Comments 6: Allowing clinicians to choose the changeover method (line 140) introduces potential selection bias; this should be acknowledged more directly in the Limitations section.
Response 6: You are absolutely correct. We have added a note at the end of the limitations section regarding the potential existence of selection bias.
Comments 7: The ethical considerations are well described; including the IRB name and approval number would add completeness.
Response 7: As you pointed out, we have clearly stated the name of the ethics committee and the approval number.
Results
Comments 8: The demographic data (lines 168–175) could be supplemented with a summary table of key statistical differences.
Response 8: We sincerely appreciate the reviewer’s valuable suggestion to include a summary table of demographic data with key statistical differences. While we agree that such a table could provide additional clarity, we are concerned that adding another table may result in an excessive number of figures and tables in the manuscript. To maintain readability and conciseness, we have instead ensured that all key demographic characteristics and their statistical comparisons are clearly described in the text. We hope this approach will be acceptable.
Comments 9: The finding that age was the only factor associated with fluctuations (lines 212–220) is important and could be highlighted with a simple visual (bar chart or forestplot).
Response 9: Thank you for your feedback. To improve readability, Table 4 has been replaced with Figure 2, which is a forest plot. This has significantly improved the manuscript.
Discussion
Comments 10: The main findings are clearly summarized and interpreted appropriately.
The section on increased catecholamine sensitivity in older adults is interesting;
consider adding supporting literature if available.
Response 10: We sincerely thank the reviewer for the positive and encouraging comments on the Discussion section. We appreciate the insightful suggestion regarding the section on increased catecholamine sensitivity in older adults. However, we found that there are few existing studies directly addressing this specific topic, making it difficult to add appropriate supporting references. We have therefore retained the current description while noting this limitation in the Discussion.
Limitations
Comments 11: While limitations are well identified, more emphasis should be placed on the potential bias from allowing staff to select the changeover method.
Response 11: As stated above, We have added a note at the end of the limitations section regarding the potential existence of selection bias.
Conclusion
Comments 12: The conclusion is clear. Suggesting future multicenter or prospective studies would strengthen it.
Response 12: We added the expression “Future multicenter and prospective studies are warranted to validate these findings” at the end of the conclusion.
References
Comments 13: The references are relevant but somewhat limited in scope. Expanding to include recent studies (2021–2025) and clinical guidelines would better position the work within current evidence.
Ensure consistency in formatting, inclusion of DOIs, and removal of residual template text (e.g., “Nurs. Rep. 2025, 15, x FOR PEER REVIEW 12 of 12”).
Check that all references are cited in the text in correct sequence and that there are no duplicates.
This is a well-executed study on an important topic. The methodology is robust,
and the findings are clinically relevant. Addressing the points above—particularly clarifying the title, explicitly noting the retrospective design, improving presentation of results, and expanding the reference list—would further strengthen the manuscript.
Response 13: Thank you for your feedback. As mentioned above, prior research in this field is limited, making it difficult to add additional references. Finally, we have verified that there is no duplication in the paper.
Reviewer 3 Report
Comments and Suggestions for Authors
First of all, I recommend publication of this interesting article. I have no concerns with regard to the quality of this research. In addition, the authors make careful conclusions and describe the limitations of this retrospective study adequately. During my work as an anesthesist in the operating theatre, I also made the experience that old patients generally require higher doses of catecholamines during operations. Consequently, these emergency department and critical care unit data make sense to me.
My only concern is the general practice of catecholamine syringe changeover techniques and I recommend to at least consider the following practice to enhance patients' safety. When I used to work as an anesthesist and had highly unstable patients during an operation, I applied double pumping changeover and did not experience any problems with it. I am therefore surprised that this was described as an error-prone method that can lead to hypotension. The way I applied it was to have 2 perfusor pumps. Rather than exchanging one pump to another by switching the 3 way stopcock (as described in the article), both pumps were continuously running at the same time and continuously connected to the 3 way stopcock. For example, one perfusor pump was running at a rate of 9.9 ml per hour and the other was running at a rate of 0.1 ml per hour, both being connected to a 3 way stopcock and both running all the time during the operation. When there was an alarm that the 9.9 ml per hour pump will be empty in 3 minutes, I first increased the rate of the 0.1 ml per hour perfursor to 1.1 ml per hour, at the same time reducing the rate of the 9.9 ml per hour pump to 8.9 ml per hour, then I increased the 1.1 ml/h perfusor to 2.1 ml/h and reduced the 8.9 ml/h perfusor to 7.9 ml/h and so on, until I had again one full perfusor running at a rate of 9.9 ml/h and the other (almost empty) perfusor running at 0.1 ml/h. Then I could easily exchange the (almost empty) 0.1 ml/h perfusor without observing hemodynamic instability. Importantly, both perfusors were constantly running all the time during the operation, just at different rates. The change of perfusors was only done in the perfusor with the very low rate (i.e. the 0.1 ml/h rate). This changeover method may take a few minutes extra time, but I am convinced that it is worth investing this time for hemodynamically unstable patients.
Author Response
Comment: My only concern is the general practice of catecholamine syringe changeover techniques and I recommend to at least consider the following practice to enhance patients' safety. When I used to work as an anesthesist and had highly unstable patients during an operation, I applied double pumping changeover and did not experience any problems with it. I am therefore surprised that this was described as an error-prone method that can lead to hypotension. The way I applied it was to have 2 perfusor pumps. Rather than exchanging one pump to another by switching the 3 way stopcock (as described in the article), both pumps were continuously running at the same time and continuously connected to the 3 way stopcock. For example, one perfusor pump was running at a rate of 9.9 ml per hour and the other was running at a rate of 0.1 ml per hour, both being connected to a 3 way stopcock and both running all the time during the operation. When there was an alarm that the 9.9 ml per hour pump will be empty in 3 minutes, I first increased the rate of the 0.1 ml per hour perfursor to 1.1 ml per hour, at the same time reducing the rate of the 9.9 ml per hour pump to 8.9 ml per hour, then I increased the 1.1 ml/h perfusor to 2.1 ml/h and reduced the 8.9 ml/h perfusor to 7.9 ml/h and so on, until I had again one full perfusor running at a rate of 9.9 ml/h and the other (almost empty) perfusor running at 0.1 ml/h. Then I could easily exchange the (almost empty) 0.1 ml/h perfusor without observing hemodynamic instability. Importantly, both perfusors were constantly running all the time during the operation, just at different rates. The change of perfusors was only done in the perfusor with the very low rate (i.e. the 0.1 ml/h rate). This changeover method may take a few minutes extra time, but I am convinced that it is worth investing this time for hemodynamically unstable patients.
Response: Thank you for your feedback. Adverse events likely do not occur when syringe exchanges are performed by experienced physicians like you. This study includes less experienced nurses and aims to provide evidence for clinical practice that can be applied in settings where nurses do most of the catecholamine syringe exchanges. The method you shared seems groundbreaking. We will consider it as a topic for future research. We sincerely thank you for dedicating your valuable time to reviewing this study.
Round 2
Reviewer 2 Report
Comments and Suggestions for Authors I suggest that the paper be published < !--a=1-->< !--a=1-->Author Response
We sincerely thank you for reviewing our paper. Thanks to your valuable comments, the quality of the manuscript has been significantly improved. We greatly appreciate your contribution.